# An In Silico Molecular Modelling-Based Prediction of Potential Keap1 Inhibitors from *Hemidesmus indicus* (L.) R.Br. against Oxidative-Stress-Induced Diseases

**DOI:** 10.3390/molecules28114541

**Published:** 2023-06-03

**Authors:** Senthilkumar Vellur, Parasuraman Pavadai, Ewa Babkiewicz, Sureshbabu Ram Kumar Pandian, Piotr Maszczyk, Selvaraj Kunjiappan

**Affiliations:** 1Department of Biotechnology, Kalasalingam Academy of Research and Education, Krishnankoil 626126, India; v.senthilkumar@klu.ac.in (S.V.); srkpandian@klu.ac.in (S.R.K.P.); 2Department of Pharmaceutical Chemistry, Faculty of Pharmacy, M.S. Ramaiah University of Applied Sciences, Bengaluru 560054, India; pvpram@gmail.com; 3Department of Hydrobiology, Faculty of Biology, University of Warsaw, 02-089 Warsaw, Poland; ewa.babkiewicz@cnbc.uw.edu.pl; 4Biological and Chemical Research Centre, University of Warsaw, 02-089 Warsaw, Poland

**Keywords:** oxidative stress, *Hemidesmus indicus* (L.) R.Br., Keap1–Nrf2, molecular docking, antioxidant

## Abstract

The present study investigated the antioxidant potential of aqueous methanolic extracts of *Hemidesmus indicus* (L.) R.Br., followed by a pharmacoinformatics-based screening of novel Keap1 protein inhibitors. Initially, the antioxidant potential of this plant extract was assessed via antioxidant assays (DPPH, ABTS radical scavenging, and FRAP). Furthermore, 69 phytocompounds in total were derived from this plant using the IMPPAT database, and their three-dimensional structures were obtained from the PubChem database. The chosen 69 phytocompounds were docked against the Kelch–Neh2 complex protein (PDB entry ID: 2flu, resolution 1.50 Å) along with the standard drug (CPUY192018). *H. indicus* (L.) R.Br. extract (100 µg × mL^−1^) showed 85 ± 2.917%, 78.783 ± 0.24% of DPPH, ABTS radicals scavenging activity, and 161 ± 4 μg × mol (Fe (II)) g^−1^ ferric ion reducing power. The three top-scored hits, namely Hemidescine (−11.30 Kcal × mol^−1^), Beta-Amyrin (−10.00 Kcal × mol^−1^), and Quercetin (−9.80 Kcal × mol^−1^), were selected based on their binding affinities. MD simulation studies showed that all the protein–ligand complexes (Keap1–HEM, Keap1–BET, and Keap1–QUE) were highly stable during the entire simulation period, compared with the standard CPUY192018–Keap1 complex. Based on these findings, the three top-scored phytocompounds may be used as significant and safe Keap1 inhibitors, and could potentially be used for the treatment of oxidative-stress-induced health complications.

## 1. Introduction

Oxygen is an essential element in the respiratory processes of most living cells [1]. Cells utilize oxygen via the breakdown of biomolecules to generate energy in the form of ATP (adenosine triphosphate), and free radicals form as unavoidable by-products in the mitochondria [2]. Generally, the produced free radicals are highly reactive atoms with one or more unpaired electrons in their external shell. Free radicals can quickly lose or gain a single electron, acting as powerful oxidants or reductants [3]. These free radicals are generally termed reactive oxygen species (ROS), reactive nitrogen species (RNS), and DNA-reactive aldehyde (DRA) [4]. ROS is a double-edged sword with functional (lower level) and harmful effects (higher level) [5]. At lower levels, ROS is essential for protein phosphorylation, transcription factor activation, cell cycle, cell division, cellular proliferation, migration, and programmed cell death, as well as some metabolic functions, whereas higher levels of ROS cause oxidative damage and lead to irreparable damage of the cell [6]. Accumulative evidence indicates that oxidative stress has been linked to numerous pathological consequences, including cancer, diabetes mellitus, cardiovascular diseases, inflammation, aging, and neurodegenerative diseases (Alzheimer’s disease, epilepsy, and Parkinson’s disease) [7,8].

The aim is to maintain the level of ROS in cells by scavenging free radicals through several enzymatic and non-enzymatic antioxidant cellular defense mechanisms [9]. The detoxifying enzymes and non-enzymatic antioxidants play a significant role in protecting cells, organs, and tissues from toxins and oxidative stress [10]. Oxidative-stress-responsive genes sensitive to secreted primary antioxidant enzymes against free radicals include superoxide dismutase (SOD), catalase (CAT), glutathione peroxidase (GPx), and peroxiredoxins [11]. Conversely, non-enzymatic antioxidants (carotenoids, vitamins A, C, and E, flavonoids, alpha-lipoic acid glutathione) interrupt free-radical chain reactions [12]. In addition, intracellular ROS levels elevated above a certain threshold result in the downregulation of cellular antioxidant pathways and enzyme systems, which, in turn, leads to the development of numerous complications via a variety of molecular targets, including nuclear factor-B (NF-B), nuclear factor E2 (erythroid-derived 2)-related factor 2 (Nrf2), Kelch-like ECH-associated protein 1 (Keap1), mitogen-activated protein kinases (MAPKs), and phosphoinositide 3-kinase (PI3K) [13,14,15].

Nrf2 is a potential transcriptional activator that coordinates basal and stress-inducible activation of numerous cytoprotective genes against oxidative stress [16]. Target genes of Nrf2 are involved in drug transport, xenobiotic metabolism, glutathione production, and ROS removal [17]. Keap1 is a cytoplasmic protein, and it is a predominant negative regulator of Nrf2 [18]. In a Keap1-dependent manner, Nrf2 is continuously destroyed under basal conditions via the ubiquitin–proteasome pathway [19]. By preventing the ubiquitin–proteasome system from degrading Nrf2, the inhibition of Keap1 activity can accumulate newly synthesized Nrf2 and cause it to move to the nucleus, where it triggers the transcription of several antioxidative and cytoprotective genes, ultimately activating the cell defense system [20].

Plenty of studies have indicated that plant-derived phytocompounds confer effective free-radical scavenging activities that may, in turn, reduce the ROS-induced oxidative stress and enhance antioxidative defense mechanisms [21]. In addition, plant-derived antioxidants are more preferable to synthetic ones because synthetic antioxidants have negative health consequences when used for more extended periods of time [22]. In recent years, there has been a lot of interest in Indian traditional medicinal systems and the use of plant-derived drugs as an alternative option to treat various complications including cancer, diabetes, and cardiovascular and neurodegenerative diseases, etc. [23]. *Hemidesmus indicus* (L.) R.Br. belonging to the Asclepiadaceae family often known as “Indian Sarsaparilla”, is a twining shrub that has been used as a folk medicine and as an ingredient in Siddha, Ayurvedic, and Unani medicines against inflammation, blood, and other ailments [24]. Several studies have established that the phytochemical profile of the root of *H. indicus* (L.) R.Br. extract contains 2-hydroxy-4-methoxy-benzoic acid, b-sitosterol, α- and β-amyrins, nerolidol, caryophyllene, borneol, lupeol, tetracyclic triterpene alcohols, hemidesminin, hemidesmin-1 and -2, resin acids, fatty acids, tannins, glycosides, and ketone, etc., which are used for the treatment of different cancers, such as leukemia, breast, hepatic, colon, and skin cancer [25]. Hence, the present study was designed to evaluate the in vitro free-radical quenching potential of a crude extract of *H. indicus* (L.) R.Br. Furthermore, the study was extended to predict potential Keap1 inhibitors from target plant *H. indicus* (L.) R.Br. to neutralize the excessive ROS generation through in silico molecular docking and dynamics simulation tools. In addition, the pharmacokinetic and physicochemical properties of the screened phytocompounds were also studied.

## 2. Results

### 2.1. Antioxidant Power

#### 2.1.1. DPPH Radical Scavenging Assay

Figure 1a shows the free-radical scavenging effect of varying concentrations of *H. indicus* (L.) R.Br. extract on DPPH radicals. The observed results show varied concentrations of 100, 75, 50, 25, 12.5, and 6.25 µg × mL^−1^
*H. indicus* (L.) R.Br. extract displayed 85 ± 2.917%, 67.667 ± 1.416%, 42.643 ± 1.335%, 34.897 ± 1.096%, and 24.523 ± 2.159%, respectively, in a concentration-based manner. Furthermore, 6.25 µg × mL^−1^ of rutin showed 34.72 ± 3.999% scavenging of DPPH radicals.

#### 2.1.2. ABTS Radical Scavenging Assay

The standard drug has been used to compare the relative antioxidant effects to scavenge the ABTS radical. Using potassium persulphate, the stable form of the ABTS radical cation was generated. After producing stable absorbance, the reaction medium is supplemented with an antioxidant *H. indicus* (L.) R.Br. extract, and the antioxidant power is measured by observing decolourization. The varied concentrations of 100, 75, 50, 25, 12.5, and 6.25 µg × mL^−1^ of *Hemidesmus indicus* extract revealed ABTS radical ions in a concentration-dependent mode, as shown in Figure 1b. At the maximum concentration (100 µg × mL^−1^) of aqueous methanolic extract of *H. indicus* (L.), R.Br. exhibited the highest (78.783 ± 0.24%) decolourization of ABTS radicals.

#### 2.1.3. FRAP Assay

When an *H. indicus* (L.) R.Br. extract (antioxidant) reacts with a ferric tripyridyltriazine (Fe^3+^ TPTZ) complex to produce a coloured ferrous tripyridyltriazine (Fe^2+^ TPTZ), the FRAP assay determines the reducing potential of the ferric ions. The free-radical chain breaking takes place by donating an electron. Varied concentrations of aqueous methanolic extract of *H. indicus* (L.) R.Br. extracts were screened via FRAP assay along with standard ascorbic acid. In the results obtained, better reducing potential was exhibited at 161 ± 40 μg × mol^−1^ (Fe(II)) g^−1^ at a concentration of 100 μg × mL^−1^ of *H. indicus* (L.) R.Br. extract. In the observed results, *H. indicus* (L.) R.Br. extract showed higher activity than the reference standard ascorbic acid, as shown in Figure 1c.

### 2.2. Active Compounds Library

The sixty-nine identified active compounds from *H. indicus* (L.) R.Br. along with the standard drug CPUY192018 were listed in Table 1. Selected compound structures were optimized and used in silico molecular docking against the chosen protein, the Kelch–Neh2 complex.

### 2.3. Active Binding Site Identification

The predicted active binding sites of the target protein (Kelch–Neh2 complex) are presented in Figure 2. The Prank Web tool revealed that there are three important binding sites where the active ligands interact and trigger conformational changes. The predicted binding pockets were presented in three different colours (blue, red, and green). The first binding pocket (blue colour) is the highest score of all three pockets, with a pocket score of 5.91, 18 amino acids, and a probability score of 0.310. The second binding pocket (red colour) with a pocket score of 4.11, 14 amino acids, and the probability score was 0.176. Additionally, the third binding pocket (green colour) had the least pocket score of 1.21, 8 amino acids, and a probability score was 0.012. The structure-based molecular screening of the selected phytocompounds was evaluated using the same identified binding pockets of the Keap1 protein. In the molecular docking studies, receptor grid construction resulted in more reliable ligand posture scoring. As a result, based on the previously acquired binding site residues, we constructed a receptor grid for the selected Kelch–Neh2 complex protein in order to obtain a more exact scoring of our ligand-binding process. The receptor grid with a box dimension of X = 64.30 Å, Y = 53.75 Å, and Z = 48.41 Å was created and used for molecular docking analysis. The receptor, the Keap1 protein grid box, is displayed in Appendix A.

### 2.4. Molecular Docking

The intermolecular interaction between the target protein and active compounds was investigated using in silico molecular docking method. Using the PyRx tool in the AutoDock Vina program, a certain number of bioactive compounds (sixty-nine) and a standard drug CPUY192018 were docked against the Kelch–Neh2 complex protein to analyze their binding potential. Ten bioactive compounds were shown to have a significant binding affinity (>−9.00 Kcal × mol^−1^) against the target Kelch–Neh2 complex protein. Following molecular docking investigations, it was discovered that the bioactive compounds’ binding energies were dispersed, ranging from −4.60 to −11.30 Kcal × mol^−1^, as shown in Figure 3 and Table 1. The top three compounds (Hemidescine (−11.30 Kcal × mol^−1^), Beta-Amyrin (−10.00 Kcal × mol^−1^), and Quercetin (−9.80 Kcal × mol^−1^)) were chosen for future research based on their binding energy with the amino acid residues in the active site of the Kelch–Neh2 complex protein. Along with this study, we used a standard drug CPUY192018 (−9.10 Kcal × mol^−1^).

### 2.5. Interpretation of Receptor–Ligand Interactions

Hemidescine (HEM) created three hydrophobic contacts, ALA366 (3.77 Å), VAL514 (3.65 Å), VAL561 (3.80 Å), whereas GLY367 (2.89 Å), ARG415 (2.67 Å), ILE416 (2.32 Å), VAL418 (2.76 Å), VAL561 (3.26 Å), VAL561 (2.68 Å), GLN563 (3.33 Å) formed seven hydrogen bonds and one salt bridge ARG326X (4.85 Å) with the target protein, as presented in Figure 4a,b. Beta-Amyrin (BET) formed four hydrophobic interactions with ALA336 (3.90 Å), VAL418 (3.44 Å), VAL514 (3.51 Å), VAL561 (3.93 Å) target protein, as depicted in Figure 4c,d). Quercetin (QUE) formed one hydrophobic bond, ALA366X (3.93 Å), and eight hydrogen bonds, GLY367 (2.14 Å), ARG415 (1.89 Å), ARG415 (2.30 Å), VAL465 (2.31 Å), VAL465 (2.44 Å), ALA510 (2.08 Å), VAL512 (2.72 Å), VAL606 (2.39 Å), with the target protein, as depicted in Figure 4e,f. Standard drug CPUY192018 (CPU) formed one hydrophobic bond, VAL514 (3.66 Å), and five hydrogen bonds, GLY367 (2.46 Å), VAL418 (1.55 Å), VAL465 (1.97 Å), VAL561 (2.09 Å), and VAL608 (3.26 Å), with the target protein, as depicted in Figure 4g,h. The list of bonding interactions between the three selected bioactive compounds and standard drug CPUY192018 with Keap1 protein is presented in Appendix A.

### 2.6. In silico Prediction of Physicochemical and ADME Properties

The pharmacokinetic (ADME) and physicochemical properties of the phytocompounds from *H. indicus* (L.) R.Br. were assessed using the SwissADME online tool (http://www.swissadme.ch/, accessed on 3 October 2022); the observed results are presented in Table 2. The molecular weights of the top-scored phytocompounds, Hemidescine, Beta-Amyrin, Quercetin, and standard drug CPUY192018 are 650.84.38, 426.72, 302.24, and 614.64 g × mol^−1^, respectively, as shown by the data in Table 2, which was determined to be contrary to Lipinski’s rule of five because the molecular weights of all top-scored phytocompounds and standard drug are higher (MW > 350). The polar surface areas of the three top-scored phytocompounds, Hemidescine, Beta-Amyrin, Quercetin, and standard drug CPUY192018 are 131.14 Å^2^, 20.23 Å^2^, 131.36 Å^2^, and 184.58 Å^2^, respectively. The predicted results also demonstrated a low rate of gastrointestinal (GI) absorption in humans for the bioactive substances Hemidescine and Beta-Amyrin, whereas Quercetin showed better gastrointestinal (GI) absorption. The higher the number of H-bonds, the more likely they are engaged in protein–ligand interactions. Hemidescine, Beta-Amyrin, and Quercetin are the phytocompounds found in the target plant, and are therefore more likely to develop into a drug-like candidate with potential Keap1–Kelch inhibitors. The phytocompounds Hemidescine, Beta-Amyrin, and Quercetin were shown to have synthetic accessibility scores of 8.01, 6.04, 3.23, respectively, indicating that they are challenging to synthesize.

Figure 5 shows the bioavailability radar plots of the top-scored phytocompounds Hemidescine, Beta-Amyrin, and Quercetin, as well as standard CPUY192018, and their drug-like properties. The pink zone within the hexagon signifies the optimal range for the compounds. The drug-like compound’s recommended range was insaturation (INSITU): fraction of carbons in the sp3 hybridization no less than 0.25, insolubility (INSOLE): log S no higher than 6, hydrophobicity (LIPO): between −0.7 and +5.0, rotatable bonds (FLEXI): no more than 9 rotatable bonds, molecular weight (SIZE): between 150 and 500 g × mol^−1^, polar surface area (POLAR): between 20 and 130 g × mol^−1^ and polar surface area (POLAR): between 20 and 130 Å^2^. The red slanted hexagon off-shoot of the vertex displays drug-like properties of the phytocompounds Hemidescine, Beta-Amyrin, and Quercetin and standard CPUY192018 (Figure 5). Moreover, using a BOILED-Egg model, the pharmacokinetic characteristics of the top-scored phytocompounds, Hemidescine, Beta-Amyrin, Quercetin and standard CPUY192018 were examined. The BOILED-Egg model proved helpful in predicting two important pharmacokinetic characteristics at the same time, namely, passive gastrointestinal absorption (HIA) and blood–brain barrier (BBB) penetration. The chemical in the yolk (i.e., yellow area) indicates BBB permeation very likely, whereas albumin (i.e., white region) represents highly possible HIA absorption in the egg-shaped organization plot. The bioactive components, Hemidescine, Beta-Amyrin, and standard CPUY192018 were detected outside the BOILED-Egg model in Figure 6, indicating poor gastrointestinal absorption, whereas Quercetin was detected in the white region, indicating higher gastrointestinal absorption. The top-scored phytocompounds Hemidescine, Beta-Amyrin, and Quercetin have significant potential to be drug-like agents for Keap1–Kelch inhibitors and are used for oxidative-induced diseases, as indicated by the above-mentioned expected findings.

### 2.7. Analysis of Toxicity

The in silico toxicity prediction of the top-scoring phytocompounds, Hemidescine, Beta-Amyrin, Quercetin, and the standard drug CPUY192018, was evaluated using the pkCSM-pharmacokinetics web-based platform. The results of the top-scoring compounds of *H. indicus* (L.) R.Br. were shown in Table 3 for the predictions for AMES toxicity, drug-induced hERG toxicity, LD_50_ (median fatal dosage), hepatotoxicity, skin sensitization, Tetrahymena pyriformis (TP) toxicity, and minnow toxicity. According to the findings, there were no adverse effects, such as hepatotoxicity, carcinogenicity, or skin sensitization, for the bioactive substances Hemidescine and Beta-Amyrin. The instant or acute toxicity of compounds that were found to be the most effective during the exploration are specified by LD_50_.

### 2.8. Molecular Dynamics (MD) Simulation

The molecular docking calculations were followed by MD simulation of the Hemidescine, Beta-Amyrin, Quercetin, and the standard drug CPUY192018 combined with the Kelch–Neh2 complex protein. The MD trajectory events of the HEM—Kelch–Neh2 complex exposed that the protein RMSD fluctuated between 3.4 Å and 3 Å (Figure 7a). Ligand RMSD was stable, and fluctuations were between 3.2 and 2 Å. Smaller RMSD fluctuations show the stability of the HEM-binding Kelch–Neh2 complex. The protein–ligand contacts of the HEM–Kelch–Neh2 complex revealed that amino acid residue ARG326X (84%), VAL561X (58%), LEU365X (32%), ILE416X (97%), ARG415X (65%), and VAL606X (20%) contributes maximum interaction with Hemidescine for its activity (Figure 8a,b). The MD trajectory events of the BET–Kelch–Neh2 complex showed that protein RMSD fluctuated initially up to 2.5 Å and retained its stability thereafter till the end at 1.2 Å (Figure 7b). Ligand RMSD was stable, and fluctuations were between 3.2 and 2.2 Å. The protein–ligand contacts of the Beta-Amyrin–Kelch–Neh2 complex revealed that amino acid residue ARG415X (41%) and GLU79P (46%) contributes maximum interaction with Beta-Amyrin for its stability and activity (Figure 9a,b). Furthermore, the MD trajectory events of the QUE–Kelch–Neh2 protein complex showed that protein RMSD fluctuated initially up to 2.25 Å and retained its stability thereafter till the end at 1 Å (Figure 7c). Ligand RMSD was stable and fluctuations were between 2.2 and 1 Å. The protein–ligand contacts of the QUE–Kelch–Neh2 protein complex revealed that amino acid residue GLY367X (46.4%), VAL606X (51%), VAL512X (44%), VAL418X (69%), VAL463 (81%), ARG415X (58%), and LEU365X (35%) contribute the maximum interaction with Quercetin for their stability and activity (Figure 10a,b). The MD trajectory events of the standard drug CPU–Kelch–Neh2 protein complex showed that protein RMSD fluctuated initially up to 2 Å and retained its stability thereafter till the end at 1 Å (Figure 7d). Ligand RMSD was stable, and fluctuations were between 2.6 and 2.2 Å. The protein–ligand contacts of the standard CPU–Kelch–Neh2 protein complex revealed that amino acid residues VAL465X (76%), VAL514X (62%), VAL418X (73%), VAL606X (81%), VAL369X (61%), VAL512X (48%), and VAL561X (40%) contribute towards maximum interaction with the standard drug CPUY192018 for their stability and bioactivity (Figure 11a,b). RMSF analysis of all complexes shows no major fluctuation due to the binding of amino acids with the key functional groups of ligands (Figure 12). Collectively, it was noted that interaction at VAL561X, VAL606X, VAL463, ILE416X, and ARG415 are the key interacting residues for the stability and inhibition of the ligands–Kelch–Neh2 protein complex. Hydroxy substitution plays a major role in interacting with key amino acids. Timeline representations (H-bonds, Hydrophobic, Ionic, water bridges) of all the amino acid residues were also observed in Figure 13. The darker lines indicated the continuous interactions with the target. These all the interactions made the protein–ligand complex stable throughout the entire duration of the MD simulation study. Our analysis shows that HEM, BET, and QUE are appropriate candidates for additional in vitro testing for Kelch–Neh2 protein inhibition, as well as a framework for future lead optimization.

### 2.9. Density Functional Theory

DFT explored the relationship between geometry and the electronic properties of chemical compounds. HOMO (highest occupied molecular orbital) is a fundamental indicator of a compound’s ability to donate electrons; the frontier-orbital energies (HOMO and lowest unoccupied molecular orbital (LUMO) of phytocompounds are crucial for biological processes. The chosen phytocompounds HEM, BET, QUE, and standard medication (CPU) had their energies and HOMO–LUMO energy gaps evaluated using the B3LYP level with the 6-311G (d, p) basis set. Table 4 displays the HOMO–LUMO diagram. The standard medication (CPU) had a HOMO–LUMO energy gap of 2.5927, whereas BET had the greatest HOMO–LUMO energy gap among the phytocompounds at 5.2460, followed by HEM at 5.0637 and QUE at 3.1135. The energy gap measurements revealed that all three selected phytocompounds are extremely stable in nature.

## 3. Discussion

Free radicals possess unpaired electrons and are closely connected to the intracellular oxidative stress, whereas antioxidants are reducing agents that limit oxidative stress via donating electrons to free radicals [26]. Free radicals are unavoidable by-products generated during the normal cellular metabolism [27]. Once developed, these radicals may cause severe damage when in contact with important cellular organelles, including nucleic acids (DNA and RNA), proteins, and the cell membrane [28]. These free radicals interact with the antioxidants, which can eventually neutralize them before they cause damage. Plants produce numerous active phytocompounds as secondary metabolites, and many of them act as antioxidants [29]. Therefore, the present study was undertaken to evaluate the antioxidant potential of aqueous methanolic extract from *H. indicus* (L.) R.Br., through DPPH, ABTS, and FRAP radical scavenging assays. The investigation further extended to predict potential Keap1 inhibitors from *H. indicus* (L.) R.Br., through structure-based in silico molecular docking tools. Keap1/Nrf2 signalling is a pathway that activates the collection of over 200 genes that are involved in antioxidant enzymes, cytoprotective genes, and xenobiotic transporters [30]. Therefore, Keap1 is an important drug target for oxidative-stress-induced chronic diseases. Here, Keap1 inhibitors are potential drug candidates for oxidative-stress-induced diseases including cancer, diabetes, and neurodegenerative diseases. Sixty-nine phytocompounds were identified from the target plant *H. indicus* (L.) R.Br. through the IMPPAT database.

DPPH and ABTS assays are commonly used to measure in vitro the antioxidant potentials of pure compounds/crude extracts [31]. The DPPH radical is a dark violet colour in the solution; when it is neutralized and transformed into DPPH-H, it loses its colour or becomes pale-yellow. There are numerous reports of plant extracts neutralizing DPPH radicals in vitro. The aqueous methanolic extract of *H. indicus* (L.) R.Br. showed scavenging of DPPH radical production in a concentration-dependent manner. A similar result was observed with the aqueous methanolic extract of *Azolla microphylla* earlier [32]. The ABTS cation radical is generated via oxidation of ABTS using potassium persulphate. This radical cation is blue in colour and absorbs light at 734 nm. The ABTS cation radicals are reduced in the presence of hydrogen-donating antioxidants, including phenols, vitamin C, and thiols [33]. In the present study, 100 µg × mL^−1^ of methanolic extract of *H. indicus* (L.) R.Br. displayed 78.783 ± 0.24% reduced-ABTS cation radicals. Furthermore, FRAP assay is a simple, fast, and cost-effective method used to measure the antioxidant potential of plant extracts [34]. The aqueous methanol extract of *H. indicus* (L.) R.Br. showed a concentration-dependent manner in FRAP. In general, phenolic and flavonoid compounds in the extract might act as antioxidants and help to scavenge the free-radical generation. The observed results indicate that the aqueous methanol extract of *H. indicus* (L.) R.Br. is a probable antioxidant agent for further drug development against oxidative-stress-induced diseases.

Furthermore, the study extended to predict significant antioxidant compounds against oxidative-stress-induced complications from the target plant *H. indicus* (L.) R.Br. via pharmacoinformatics analysis. Currently, the use of information technology to determine the binding datasets of small molecules with known receptors is a major component for lead identification processes. Pharmacoinformatics is a collection of computer-based drug design tools including structure-based screening of compounds based on their binding affinities, pharmacokinetics, and pharmacodynamic abilities [35]. Pharmacoinformatics tool helps to narrow down the synthetic and biological steps, and it has speed up the drug development processes [36]. This tool also helps to understand how phytocompound bind in the active site of receptor protein, interactions amino acids, and inhibit/stimulate a certain protein might help researchers to find treatment possibilities for certain disease conditions [7]. In general, the phytocompounds obtained from plants are safe, cytoprotective, cost-effective, reduce/neutralize the toxins, and maintain the level of ROS in cells [37]. Here, sixty-nine phytocompounds in total were selected from *H. indicus* (L.) R.Br. through the IMPPAT database. In silico molecular docking aims for the most precise prediction and interactions between the receptor and ligand molecules for potential lead discovery. All the chosen compounds docked against the Keap1 protein and noted the binding affinities between −4.6 Kcal × mol^−1^ and −11.3 Kcal × mol^−1^. Based on the binding affinities and interactions between the ligand with the target protein, three compounds, namely, Hemidescine (−11.30 Kcal × mol^−1^), Beta-Amyrin (−10.00 Kcal × mol^−1^), and Quercetin (−9.80 Kcal × mol^−1^), as well as the standard drug CPUY192018 (−9.10 Kcal × mol^−1^), were selected. Similarly, 2,2′-(naphthalene-1,4-diylbis(((4-methoxyphenyl)sulfonyl) azanediyl))diacetic acid was discovered the most potent protein−protein interaction (PPI) inhibitor of Keap1−Nrf2 through molecular binding determinants analysis of Keap1, and fluorescence polarization assay [38].

Therapeutic efficacy of the plant-derived phytocompounds mainly depends on their pharmacokinetics (ADMET), pharmacodynamics (mechanism of compounds action), and physicochemical properties (molecular weight, number of rotatable atoms, number of hydrogen bond acceptors and donors, molar refractivity, TPSA (topological polar surface area), solubility, gastrointestinal absorption, blood–brain barrier penetration, etc.), which are considered in the novel discovery process [39]. A few crucial factors enhancing dietary phytochemicals bioactivity and health promotion include their bioavailability to target cells, as well as absorption and metabolism properties of the human body [40]. Lipinski’s rule of five is most commonly used to determine whether a phytocompound has a particular pharmacological or biological action that qualifies it as a drug that can be taken orally by humans [41]. According to the rule, a compound with molecular weight (Mw) < 500 Da, calculated logP < 5, hydrogen-bond donors < 5, hydrogen-bond acceptors < 10 and TPSA < 140 Å^2^ will be further investigated as a potential drug because it may lose important properties related to its absorption, distribution, metabolism, and excretion [42]. In the present study, the top-scored hits, two compounds (Hemidescine and Beta-Amyrin) only one violation of the parameter and the third compound, Quercetin, did not violate any of Lipinski’s rule of five. While compared with standard drugs, CPUY192018 showed two parameters violated in Lipinski’s rule of five. Phytochemicals are naturally occurring substances that can be found in a variety of plants that are consumed by humans and are generally thought to be safe. Since most phytochemicals have not recognized potential for harm, the Food and Drug Administration (FDA) in the United States does not impose restrictions on their use [43].

Prior to the lead identification, the in silico toxicity prediction studies provide information on the cellular, organ and tissue toxicity profile of the compounds, which reduces cost and time as well as minimizing the rate of late-stage rejection in a drug discovery process [44]. Additionally, toxicity testing connected with pharmacokinetic and pharmacodynamic features must be studied in order for a lead compound to be approved for commercial usage [45]. The availability of potentially beneficial drugs for human use becomes delayed, because animal research has high costs and delays in drug approval [46]. Computer-aided methods for predicting the toxicity of phytocompounds are therefore recognized as useful [47]. In the present study, the toxicity results revealed that all selected top hits had no harmful effects. The LD_50_ (median fatal dosage) notified the immediate or acute toxicity of substances that were determined to be the most effective in the investigation. The stability of molecular docking results of the protein–ligand complexes was explored through molecular dynamics simulation studies. All three protein–ligand complexes were validated by interpreting the RMSD, RMSF, hydrogen bonding, and interacting amino acids of the protein with lead phytochemical complexes and were found to be stable during the simulation. The higher HOMO value denotes a molecule with a good electron donor, whereas a lower value implies a weak electron acceptor. Furthermore, a smaller energy gap between the LUMO and HOMO energies has a considerable influence on the intermolecular charge transfer and bioactivity of molecules [48]. Thus, a wide energy gap observed in the hit molecules negatively affects the electron to move from the HOMO to the LUMO, which subsequently led to a strong affinity of the inhibitor for Keap 1 protein [49]. In addition, the HOMO–LUMO gap indicates that Beta Amyrin and Hemidescine are the most reactive ligands.

## 4. Materials and Methods

### 4.1. Plant Materials and Reagents

*Hemidesmus indicus* (L.) R.Br. “Indian Sarsaparilla” was collected from the foothills of Western Ghats in and around Kilavankoil Hills (longitude: 77.5232° and latitude: 9.6383°), Virudhunagar district, India, during the early winter season. The root portion of the collected plant was separated with the help of a fine knife. Later, the separated root was repeatedly cleaned using tap water to remove soil and other debris, then dried naturally for 72 h, and then ground into a fine powder. The ground powder was screened through a 60-mesh sieve size and safely stored in a desiccator for further experiments. The fine powder was subjected to an ultrasonic-assisted extraction (UAE). Extraction was performed in an adjustable ultrasonic bath using a 5 g fine powdered sample of *H. indicus* (L.) R.Br. along with 50 mL of methanol (80%) at specified ultrasound intensity (60 W × cm^−2^), pulse cycle (0.2 s) and temperature (40 °C) for 20 min. UV–visible spectrophotometer of Shimadzu UV-1800 series, UV Probe 2.62 software, Japan, was utilized for the analysis. Rotary vacuum dryer (Rotavapor, Buchi India Pvt, Ltd., Mumbai, India) and mixer grinder (Premier Electronics Ltd., India) were also used in this study. A sensitive colorimetric scavenger 2,2-diphenyl-1-picrylhydrazyl (DPPH) and 2,2′-Azinobis (3-ethylbenzothiazoline-6-sulphonic acid) radical cation (ABTS) assay adapted were from Sigma-Aldrich, MO, USA. Methanol, 2,4,6-tripyridyl-s-triazine (TPTZ), rutin, and gallic acid were obtained from Himedia laboratories Pvt. Ltd., Mumbai, India.

### 4.2. Antioxidant Activity

#### 4.2.1. DPPH Radical Scavenging Activity

The DPPH radical scavenging activity of *H. indicus* (L.) R.Br. extract was performed using DPPH reagent (4 g DPPH dissolved in 90% methanol), as indicated by our previously published article with slight modification [50]. Briefly, 0.1 mM DPPH solution (0.1 mM) 1 mL of freshly prepared DPPH solution was added to 3 mL of different concentrations (6.25–100 mg × mL^−1^) of *H. indicus* (L.) R.Br. extract and the reference drug (rutin) diluted in 90% methanol. The control was prepared as above with 3 mL of 90% methanol instead of a sample. The mixture was kept in the dark for 30 min at room temperature. All the samples were prepared in triplicate, and the absorbance was recorded at 517 nm using a UV–visible spectrophotometer against a blank. The percentage of inhibition of DPPH was determined from the following formula: DPPH scavenging effect (% inhibition) = (A0 − A1)/A0 × 100, where A0 is the absorbance of the control reaction, and A1 is the absorbance in the presence of all of the extracted samples and reference.

#### 4.2.2. ABTS Radical Scavenging Activity

This assay depends on the capacity of the acquired extract of *H. indicus* (L.) R.Br. sample to scavenge 2,2′-azino-bis(ethylbenzthiazoline-6-sulfonic acid (ABTS) radical cation [51]. The ABTS radical cation was generated by the interaction between 7 mM of ABTS salt solution and 2.45 mM potassium per sulphate solution (1/1, *v*/*v*) held in the dark at room temperature for 12–16 h. The created ABTS solution was carefully mixed with 0.1 mL of plant extract after being diluted with 0.3 mL of methanol. After 6 min, the reaction mixture was incubated, and a UV–visible spectrophotometer was used to measure the absorbance at 734 nm, which was set to 0.700 (0.0020). Rutin in 80% methanol was used as the reference standard, and the standard curve was used to calculate the percentage of ABTS scavenging activity. ABTS scavenging effect (% inhibition) = (A0 − A1)/A0 × 100, where A0 is the absorbance of the control reaction, and A1 is the absorbance in the presence of all of the extract samples and reference. All the samples were prepared and assayed in triplicate.

#### 4.2.3. Ferric Reducing Antioxidant Potential (FRAP) Assay

The antioxidant capacity of *H. indicus* (L.) R.Br. extract was calculated spectrophotometrically via the method proposed by Benzie and Strain [52], and a technique improved by Pulido et al. [53]. The method depends on the reduction of Fe^3+^ TPTZ (colourless complex) to Fe^2+^ tripyridyltriazine (blue coloured complex) generation through electron-donating antioxidants at low pH levels. The change in absorbance at 593 nm is used to track this process. At pH 3.6, 300 mM acetate buffer (3.1 g sodium acetate, 16 mL acetic acid), 10 mM TPTZ (2,4,6-tripyridyl-s-triazine) solution in 40 mM hydrochloric acid solution, and 20 mM FeCl_3_.6H_2_O solution were used to make the FRAP reagent. A mixture of the acetate buffer (25 mL), TPTZ (2.5 mL), and FeCl_3_ was then added (2.5 mL). After 30 min of incubation at 37 °C, the newly generated FRAP solution was allowed to react with the extract from *H. indicus* (40 µL) before the absorbance was measured at 593 nm. The FeSO_4_ standard was linear between 200 and 1000 µM. Results expressed in µM Fe(II) g^−1^ dry mass were compared with a standard, ascorbic acid. All the samples were prepared and assayed in triplicate.

### 4.3. Graph Theoretical Network Analysis

The Kyoto Encyclopedia Genes and Genomes (KEGG) database and Cytoscape software version 3.7.1 used to re-construct a signalling pathway [54]. From the signalling pathway, Keap1–Nrf2 was chosen and its activities in the signalling pathway of human hepatocellular carcinoma (has05225) was analysed, the resultant signalling network pathway was presented in Figure 14. In total, there were 139 edges and 1 node presence in the built network. The measured values of degree (11), betweenness (10.5), eccentricity (1), eigen vector (6.88E-31), radiality (29.3853211), and stress (12) have revealed the threshold value of all the measurements, as well as a significant node in the network (Table 5).

### 4.4. In silico Study

#### 4.4.1. Ligand Library Preparation

Sixty-nine known phytochemicals from the *H. indicus* (L.) R.Br. was identified from the Indian Medicinal Plants, Phytochemistry and Therapeutics (IMPPAT) database [55]. The primary phytochemicals chosen were sterols, alkaloids, and flavonoids. For in silico molecular docking experiments, the three-dimensional structures (Structure Data File format) of chosen phytocompounds were drawn using ACD/Chemsketch software, v2021.2.2, Toronto, Ontario, M5C 1B5, Canada and energy-optimized through MMFF94 (Merck molecular force field 94), and a ready-to-dock library was prepared in BIOVIA discovery studio software packages. The standard Keap1 protein inhibitor, CPUY192018 (PubChem CID: 73330369) was used to compare the inhibitory effect of the selected plant-based phytocompounds. Selected ligands were energy minimized using Avogadro tool (https://avogadro.cc/ accessed on 5 August 2022) and they were converted to .pdbqt file format using the Open Babel tool (https://openbabel.org/ accessed on 5 August 2022).

#### 4.4.2. Target Protein Preparation

Based on the graph theoretical network analysis result, the Kelch–Neh2 complex protein was selected in this study. Thus, the three-dimensional X-ray crystal structure of the Kelch–Neh2 complex protein (PDB entry ID: 2flu, resolution 1.50 Å) was retrieved from the RCSB PDB (Research Collaboratory for Structural Bioinformatics, Protein Data Bank, https://www.rcsb.org/ accessed on 5 August 2022) website [56]. Using the Swiss-PDB Viewer v4.1.0, incorrect bonds and side chain anomalies were fixed, and missing residues were added to the recovered protein [57]. The file was assigned the name target.pdb which was then saved for further investigation. Additionally, we defined the protein structure and amino acid position from active sites using BIOVIA Discovery Studio Visualizer version 4.0 software (Accelrys Software Inc., San Diego, CA, USA), which was subsequently used for the molecular docking studies.

#### 4.4.3. Investigation of Protein–Ligand Interactions

##### Active Binding Site Prediction

The discovery of structure-based lead phytocompounds and the understanding of protein function are two areas where predicting ligand binding sites from specific protein structural locations has significant implications. This indicated binding region helps in the ability of lead phytocompounds to bind and establish a strong interaction with the target protein in order to provide the most effective and advantageous catalytic effects. For further research, all potential active binding sites of the targeted phytocompounds were identified using the PrankWeb (https://prankweb.cz accessed on 3 October 2022) online tool [58].

### 4.5. Molecular Docking

Using the PyRx 0.8 tool in the AutoDock Vina program, a receptor grid box was generated once the target protein active binding site was predicted [59]. The created ready-to-dock phytochemicals library was docked against the chosen Kelch–Neh2 complex protein (PDB ID: 2flu) to explore the ligand–protein interactions [60]. The grid box with a 10.0 radius across the active binding site of the predicted area was selected to assess the molecular binding affinities (Kcal × mol^−1^) using the protein (receptor) and phytocompounds (ligand) files stored in the “.pdbqt” format. The binding energy affinities of up to 10 different docking sites for each ligand were assessed using the AutoDock Vina tool (PyRx-Python Prescription 0.8, The Scripps Research Institute, 10550 North Torrey Pines Road La Jolla, CA 92037-1000 USA). After docking, phytochemicals with best and top conformation were determined based on their root-mean-square deviation (RMSD) and S-score binding affinity values. The docked complexes of the top scoring compounds were visualized using the Discovery Studio Visualizer tool to analyze the protein–ligand complex structures, with interactions (2D and 3D plots), the number of hydrogen bonds, hydrophobic bonds, and noncovalent interactions for each complex.

### 4.6. Drug-Likeness Evaluation

The evaluation of the drug-likeness, pharmacokinetics, and toxicity profile of the drug candidate is a significant step in the drug discovery process. Here, we investigated pharmacokinetics (absorption, distribution, metabolism, and excretion), toxicity, and some physicochemical properties of the top-scored phytocompounds, such as molecular weight, octanol–water partition coefficient log P (LogP), the number of hydrogen bond acceptors, hydrogen bond donors, solubility, molar refractivity, gastrointestinal absorption, and blood–brain barrier penetration using Swiss-ADME [61] and pkCSM-pharmacokinetic online web-based tools [62] of the selected phytocompounds from *H. indicus* (L.) R.Br.

### 4.7. Molecular Dynamics Simulation Studies

The DESMOND computer programme was developed by the D.E. Shaw research group to compute the molecular dynamics (MD) simulation of protein–ligand complexes (PLC) [63]. Through MD simulation modelling, the potential effects of PLC at target binding sites in physiological situations are emphasized (Academic license, Version 2020-1, Schrödinger, LLC, New York, NY, USA, 2021-4, Schrödinger Release: QikProp). At this initial stage, the panel enables us to construct a box (10 × 10 × 10) holding water molecules and physiological features such as pH [64]. To meet the specific requirements of the study procedure, Na^+^ or Cl^−^ ions can be introduced if the pH is not present or if it needs to be raised or lowered. The TIP3P water solvation model was used to solve the docked protein–ligand complexes. The physiological salt concentration was maintained at 0.15 M while the solvated system was neutralized using counter ions. The OPLS AA (Optimal Potentials for Liquid Simulation—All Atom) force field was applied to the PLC system [65]. A moderate minimization is performed at roughly 100 ps on the ready PLC with the help of the system builder panel. As a result, the prepared system stabilizes in response to its surroundings. The Reversible Reference System Propagator Algorithms (RESPA) integrator [66], Martyna–Tobias–Klein barostat, and the Nose–Hoover chain thermostat were all employed in molecular dynamics with two ps relaxation durations [67]. The final version of the MD simulation was created using the equilibrated system. The MD simulation was run for 100 ns at 310.15 K temperature and 1.0 bar pressure using the NPT (Isothermal–Isobaric ensemble, constant temperature, constant pressure, and constant number of particles) ensemble with the default relaxation parameters [68]. Once the simulation is complete, the findings are analyzed using a simulation interaction diagram [54].

### 4.8. Density Functional Theory (DFT)

The electronic properties of the drug-like compounds played a crucial role in its pharmacological activity. The three-dimensional electronic density system may be used to measure the electronic states of atoms, molecules, and solids using DFT theory. DFT’s primary objective is to use the basic principles of quantum mechanics to create a quantitative knowledge of material qualities. The Gaussian 03W software and the GaussView molecular visualization tools were used for computational calculations in this work to determine the top-scored binding bioactive chemicals acquired from molecular docking. The molecular structures of the chosen bioactive compound were optimized via the DFT/Becke-3-Lee-Yang-Parr (B3LYP) method using a 6-311G (d.p) basis set. Using the optimized structures, the frontier molecular orbital energies of the chosen bioactive compounds were determined, including their energy gaps (Eg) and lowest unoccupied molecular orbitals (ELUMO) and highest occupied molecular orbitals (EHOMO). GaussView, a molecular visualization programme, was used to depict the molecular orbital energy diagrams that were produced for the chosen bioactive chemicals [69].

## 5. Conclusions

Plants and plant-based products consist of an array of antioxidant phytocompounds crucial for preventing and treating many chronic diseases. Nrf2 is a central transcription factor in oxidative stress responses and is connected to numerous chronic diseases; it is activated after dissociating from Keap1. This study explored the quantification of free-radical scavenging potential from *H. indicus* (L.) R.Br. aqueous methanolic extract and found significant Keap1 inhibitors from this target plant via in silico molecular docking and molecular dynamics simulation studies. Through a thorough molecular docking investigation of phytocompounds, the three top-scored phytocompounds (Hemidescine, Beta-Amyrin, and Quercetin) were selected on the basis of least/better binding affinities than standard Keap1 inhibitor (CPUY192018). In addition, the molecular dynamics simulation studies exhibited that the three top-scored compounds with Keap1 protein complexes were highly stable during the entire simulation period. Moreover, pharmacokinetic and physicochemical properties confirm the drug-likeness and safety profiles of three selected phytocompounds. These studies further confirm antioxidant activity and effective inhibition of Keap1 protein against oxidative-stress-induced health complications. Additional in vitro and in vivo animal studies are necessary to evaluate the Keap1 protein inhibition and Nrf2 protein activation of these phytocompounds.

## Figures and Tables

**Figure 1 molecules-28-04541-f001:**
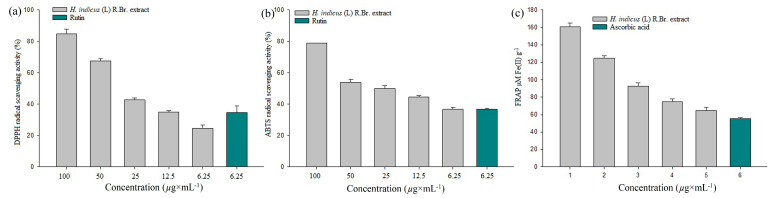
DPPH radical scavenging activities of various concentrations of aqueous methanolic extract of *H. indicus* (L.) R.Br. (**a**); ABTS radical scavenging activities of various concentrations of aqueous methanolic extract of *H. indicus* (L.) R.Br. (**b**); FRAP potential of various concentration of aqueous methanolic extract of *H. indicus* (L.) R.Br. (**c**).

**Figure 2 molecules-28-04541-f002:**
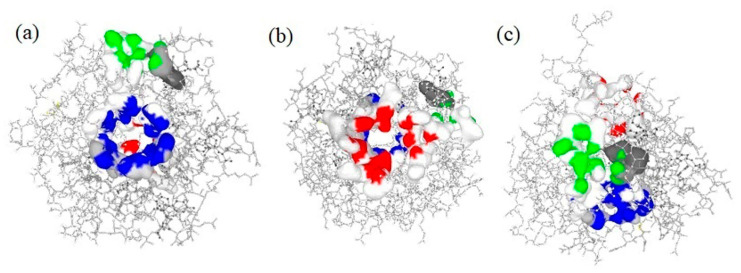
The PrankWeb tool is used to predict the binding pockets and correspondence binding sites of the Keap1 (Kelch–Neh2 complex) protein. Three binding pockets were predicted with different colours (blue, red, and green). The first binding pocket (blue colour) is the highest score of all three pockets, with a pocket score of 5.91, 18 amino acids, and a probability score of 0.310 (**a**). The second binding pocket (red colour) with a pocket score of 4.11, 14 amino acids, and a probability score was 0.176 (**b**). The third binding pocket (green colour) had the least pocket score of 1.21, 8 amino acids, and the probability score was 0.012 (**c**).

**Figure 3 molecules-28-04541-f003:**
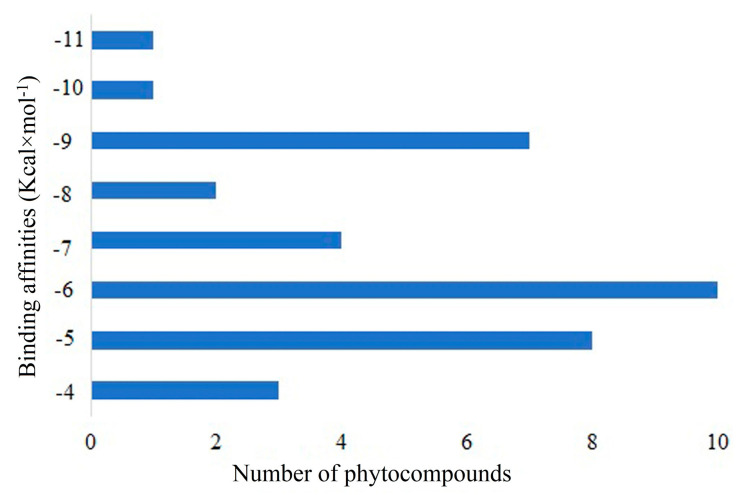
Showing the range of molecular docking score distribution of sixty-nine phytocompounds present in the *H. indicus* (L.) R.Br.

**Figure 4 molecules-28-04541-f004:**
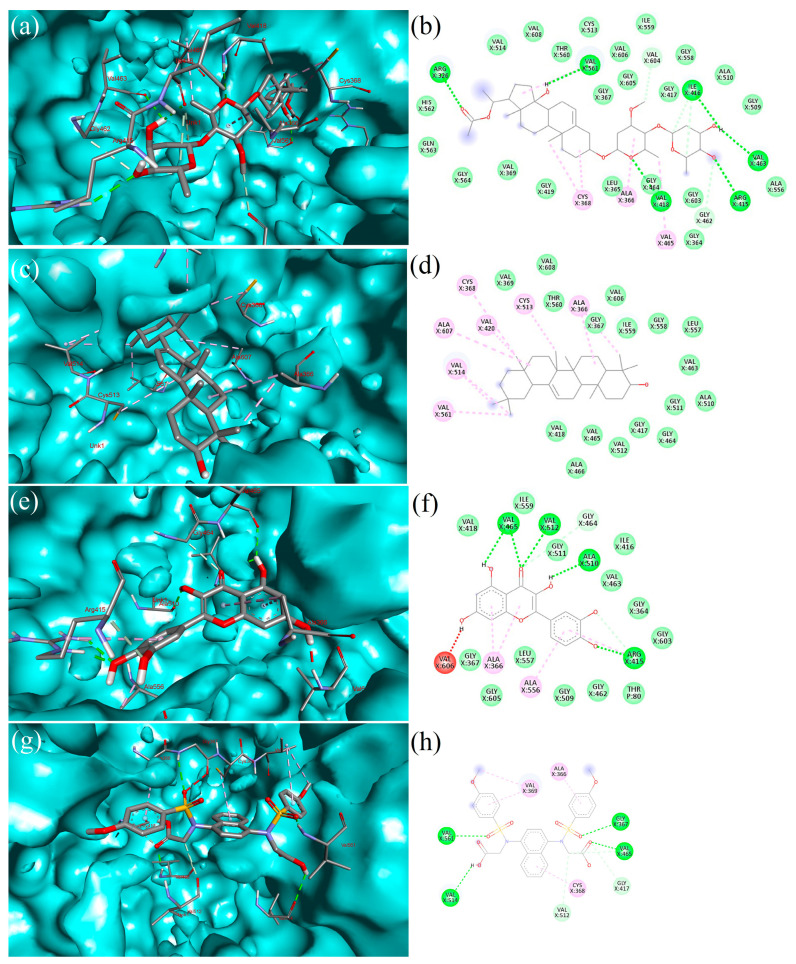
Depicted the interaction between the compound Hemidescine–Keap1 protein complex. The left side representing 3D (**a**); and the right side represents the 2D complex protein–ligand interaction (**b**); the interaction between the compound Beta-Amyrin–Keap1 protein complex. Left side representing 3D (**c**); and the right side represents the 2D complex protein–ligand interaction (**d**); the interaction between the compound Quercetin–Keap1 protein complex. The left side represents 3D (**e**); and the right side represents 2D complex protein–ligand interaction (**f**); the interaction between the standard drug (CPUY192018) and the Keap1 protein. The left side represents 3D (**g**); and the right side represents the 2D complex protein–ligand interaction (**h**).

**Figure 5 molecules-28-04541-f005:**
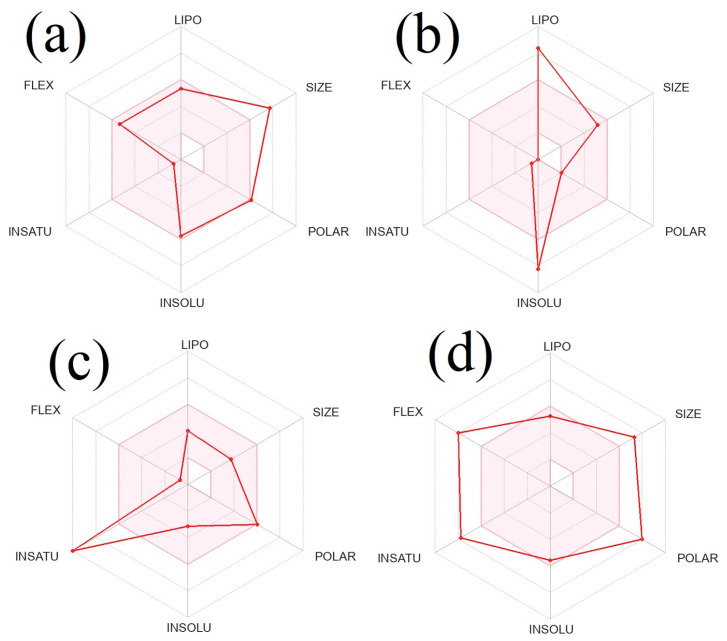
Bioavailability radar plot for oral bioavailability of top-scored phytocompounds. Hemidescine (**a**); Beta-Amyrin (**b**); Quercetin (**c**); and standard drug (CPUY192018) (**d**). The pink area exhibits the optimal range for each property (Lipophilicity as XLOGP3 between −0.7 and +5.0; Size as molecular weight between 150 and 500 g × mol^−1^; Polarity as TPSA (topological polar surface area) between 20 and 130 Å^2^; insolubility in water by log S scale no higher than 6; insaturation as per fraction of carbons in the sp3 hybridization no less than 0.25 and flexibility as per rotatable bonds no more than 9).

**Figure 6 molecules-28-04541-f006:**
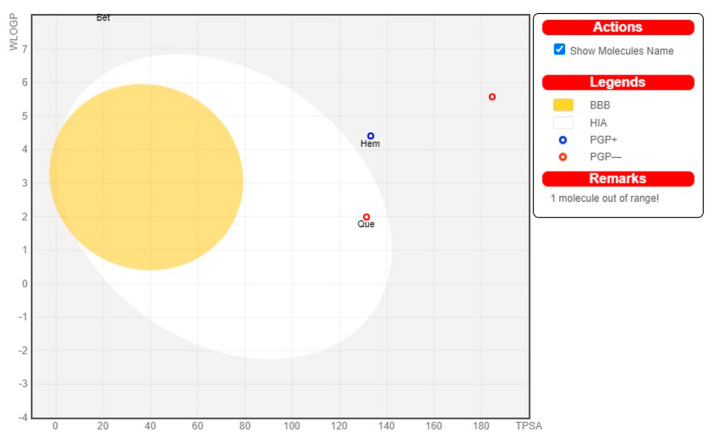
The BOILED-Egg model for the top-scored phytocompounds and standard drug CPUY192018. The BOILED-Egg represents the intuitive evaluation of passive gastrointestinal absorption (HIA) (white part) and brain penetration (BBB) (yellow part); substrates (PGP+) and non-substrates (PGP–) of the permeability glycoprotein (PGP) are represented by the blue and red colour circles, respectively, of the selected bioactive compound and standard MAPK6 inhibitor in the WLOGP-versus-TPSA graph. The grey region is the physicochemical space of compounds predicted to exhibit high intestinal absorption.

**Figure 7 molecules-28-04541-f007:**
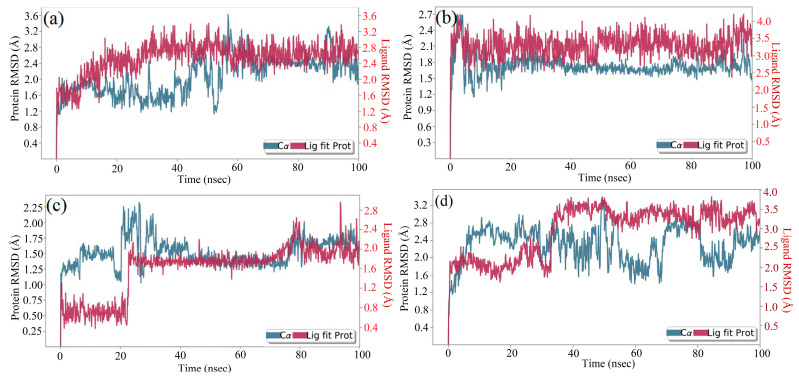
RMSD study plot for 100 ns MD Simulation of Hemidescine–Keap1 protein docked complex (**a**); Beta-Amyrin–Keap1 protein docked complex (**b**); Quercetin–Keap1 protein docked complex (**c**); and standard drug CPUY192018–Keap1 protein docked complex (**d**).

**Figure 8 molecules-28-04541-f008:**
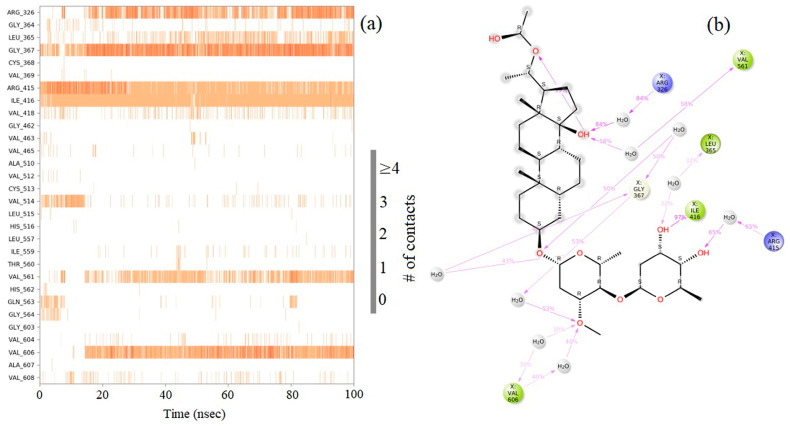
Hemidescine–Keap1 protein contacts timeline representation (**a**); and Hemidescine contacts with respect to the amino acids in the target protein Keap1 (**b**).

**Figure 9 molecules-28-04541-f009:**
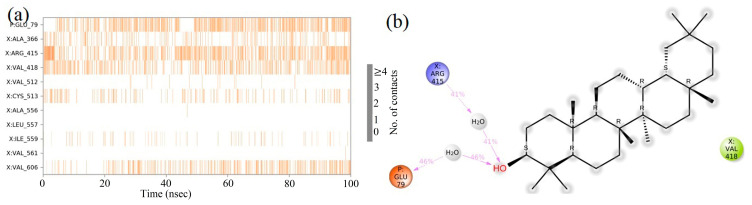
Beta-Amyrin–Keap1 protein contacts timeline representation (**a**); and Beta-Amyrin contacts with respect to the amino acids in the target protein Keap1 (**b**).

**Figure 10 molecules-28-04541-f010:**
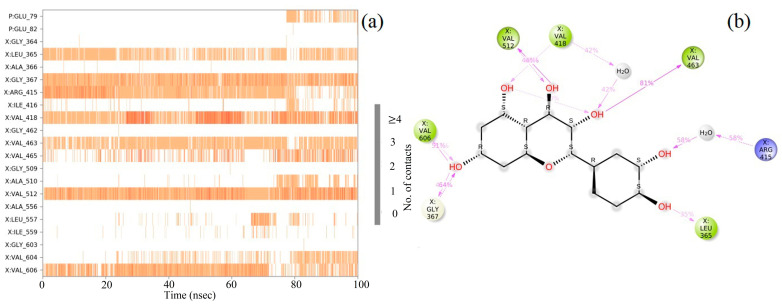
Quercetin–Keap1 protein contacts timeline representation (**a**); and Quercetin contacts with respect to the amino acids in the target protein Keap1 (**b**).

**Figure 11 molecules-28-04541-f011:**
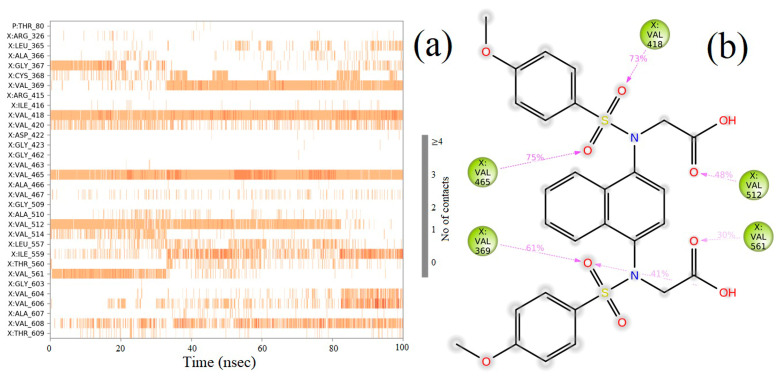
CPUY192018–Keap1 protein contacts timeline representation (**a**); and CPUY192018 contacts with respect to the amino acids in the target protein Keap1 (**b**).

**Figure 12 molecules-28-04541-f012:**
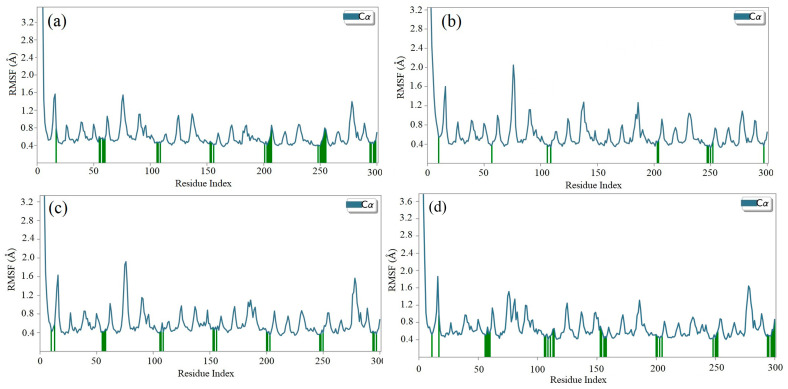
RMSF of the Hemidescine for characterizing changes in the ligand atom positions (**a**); RMSF of the Beta-Amyrin for characterizing changes in the ligand atom positions (**b**), RMSF of the Quercetin for characterizing changes in the ligand atom positions (**c**); and RMSF of the CPUY192018 (standard drug) for characterizing changes in the ligand atom positions (**d**).

**Figure 13 molecules-28-04541-f013:**
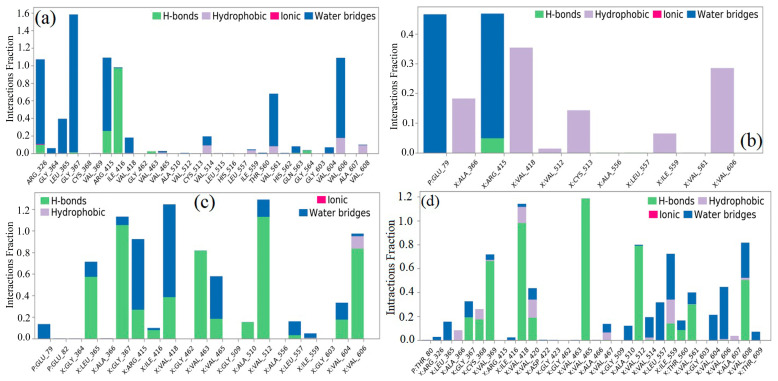
Percentage of amino acid and water mediated interactions contribution in MD simulations with Hemidescine (**a**); Beta-Amyrin (**b**); Quercetin (**c**); and standard drug (CPUY192018) (**d**).

**Figure 14 molecules-28-04541-f014:**
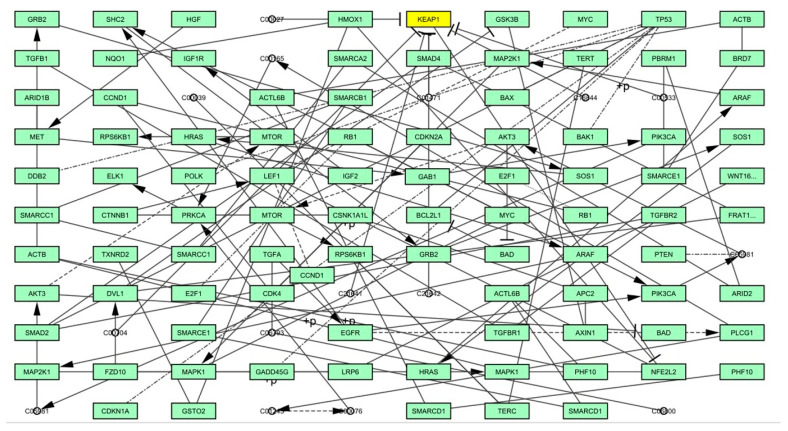
Graph theoretical network analysis of Keap1–Nrf2.

**Table 1 molecules-28-04541-t001:** Phytocompounds found in *Hemidesmus indicus* (L.) R.Br. and their binding affinity against Keap1.

S. No	Compound ID (CID)	Active Compound	Docking Score (Kcal × mol^−1^)
1.	379	Octanoic acid	−4.6
2.	643731	trans-2,cis-6-Nonadienal	−4.7
3.	2969	Decanoic acid	−4.8
4.	785	Hydroquinone	−5.0
5.	12412	Hexatriacontane	−5.0
6.	21146488	(1S,3aR,5aR,5bR,7aS,11aS,11bR,13aR,13bR)-3a,5a,5b,8,8,11a-hexamethyl-1-propan-2-yl-1,2,3,4,5,6,7,7a,9,10,11,11b,12,13,13a,13b-hexadecahydrocyclopenta[a]chrysene	−5.0
7.	460	Guaiacol	−5.1
8.	6998	Salicylaldehyde	−5.1
9.	31244	4-Methoxybenzaldehyde	−5.1
10.	3893	Lauric acid	−5.2
11.	22311	Limonene	−5.2
12.	9007	3-Methoxyphenol	−5.3
13.	985	Palmitic acid	−5.6
14.	2758	Eucalyptol	−5.7
15.	4133	Methyl salicylate	−5.7
16.	69600	2-Hydroxy-4-methoxybenzaldehyde	−5.7
17.	1183	Vanillin	−5.7
18.	444539	Cinnamic acid	−5.8
19.	8294	Linalyl acetate	−5.8
20.	12127	Isovanillin	−5.9
21.	11230	4-Carvomenthenol	−5.9
22.	61130	Myrtenal	−5.9
23.	29025	Verbenone	−5.9
24.	93046	2,10-Epoxypinane	−5.9
25.	6552009	d-Borneol	−6.0
26.	10582	Myrtenol	−6.0
27.	6989	Thymol	−6.0
28.	17100	Alpha-Terpineol	−6.0
29.	121719	Pinocarvone	−6.1
30.	5355854	Pentyl cinnamate	−6.2
31.	5284507	Nerolidol	−6.2
32.	5281522	Isocaryophyllene	−6.3
33.	2537	Camphor	−6.3
34.	3469	2,5-Dihydroxybenzoic acid	−6.4
35.	6448	Bornyl acetate	−6.4
36.	30248	Dihydrocarvyl acetate	−6.4
37.	75231	4-Methoxysalicylic acid	−6.4
38.	8468	Vanillic acid	−6.4
39.	637542	4-Hydroxycinnamic acid	−6.5
40.	6918391	Beta-Elemene	−6.6
41.	6950273	Isobornyl acetate	−6.6
42.	72	3,4-Dihydroxybenzoic acid	−6.6
43.	10742	Syringic acid	−6.6
44.	3102	Benzophenone	−6.7
45.	370	Gallic acid	−6.8
46.	111037	Alpha-Terpinyl acetate	−6.8
47.	689043	Caffeic acid	−6.8
48.	445858	Ferulic acid	−6.9
49.	91354	Aromadendrene	−6.9
50.	92812	Ledol	−7.0
51.	5369459	Phenethyl cinnamate	−7.1
52.	2345	Benzyl benzoate	−7.1
53.	442343	Levomenol	−7.3
54.	100949538	Alpha-Muurolol	−7.5
55.	442393	Beta-Selinene	−7.5
56.	5280804	Isoquercitrin	−8.8
57.	5281643	Hyperoside	−8.8
58.	9548870	Ursane	−8.9
59.	92157	Lupeol acetate	−9.0
60.	5280805	Rutin	−9.1
61.	259846	Lupeol	−9.1
62.	222284	Beta-Sitosterol	−9.3
63.	9548717	Oleanane	−9.5
64.	92156	Beta-Amyrin acetate	−9.6
65.	16129778	Tannic acid	−9.6
66.	73170	Alpha-Amyrin	−9.7
67.	5280343	Quercetin	−9.8
68.	73145	Beta-Amyrin	−10
69.	101664025	Hemidescine	−11.3
**Standard Drug**			
70.	73330369	CPUY192018	−9.10

**Table 2 molecules-28-04541-t002:** Pharmacokinetics and physicochemical parameters of selected bioactive compounds and standard drug CPUY192018.

Parameter	Hemidescine (CID: 101664025)	Beta-Amyrin (CID: 73145)	Quercetin (CID: 5280343)	CPUY192018 (CID: 73330369)
Formula	C_36_H_58_O_10_	C_30_H_50_O	C_15_H_10_O_7_	C_28_H_26_N_2_O_10_S_2_
MW (g × mol^−1^)	650.84	426.72	302.24	614.64
Num. heavy atoms	46	0	22	42
Num. arom. heavy atoms	0	12	16	22
Fraction Csp3	0.92	0.93	0.00	0.14
Num. rotatable bonds	8	0	1	12
Num. H-bond acceptors	10	1	7	10
Num. H-bond donors	3	1	5	02
Molar Refractivity	171.72	134.88	78.04	154.12
TPSA (Å^2^)	131.14	20.23	131.36	184.58
Solubility class	Moderately soluble	Poorly Soluble	Soluble	Moderately soluble
GI absorption	Low	Low	High	Low
BBB permeation	No	No	No	No
Violation of Lipinski’s rule of five	1	1	0	2
Violation of Veber rule	Yes	Yes	Yes	2
Bioavailability Score	0.55	0.55	0.55	0.11
Synthetic accessibility	8.01	6.04	3.23	3.83

**Table 3 molecules-28-04541-t003:** Toxicity profile of top-scored compounds with standard drug CPUY192018.

Compound	AMES Toxicity	Max. Tolerated Dose (Human)	hERG Inhibition	LD50	Hepatotoxicity	Carcinogenicity	Skin Sensitisation	*T. pyriformis* Toxicity	Minnow Toxicity
Hemidescine (CID: 101664025)	No	−1.41	No	2.442	No	No	No	0.286	0.714
Beta-Amyrin (CID: 73145)	No	+0.33	No	2.139	No	No	No	0.599	−2.344
Quercetin (CID: 5280343)	Yes	+0.984	No	2.251	No	No	No	0.418	2.487
CPUY192018 (CID: 73330369)	No	+0.52	No	1.841	Yes	No	No	0.286	−0.527

**Table 4 molecules-28-04541-t004:** EHOMO and ELUMO and ∆E values of selected top-scored binding compounds and standard drug CPUY192018.

Compound Name	HOMO	E_HOMO_ (ev)	LUMO	E_LUMO_ (ev)	Energy Gap (Δev)
Hemidescine	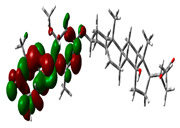	−9.5378	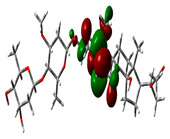	−4.4741	5.0637
Beta Amyrin	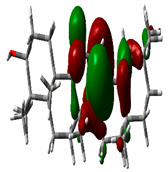	−9.8576	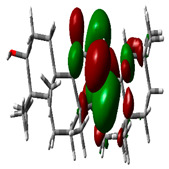	−4.6115	5.2460
Quercetin	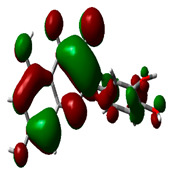	−8.4817	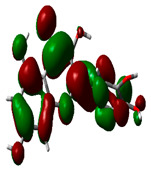	−5.3682	3.1135
Standard drug (CPUY192018)	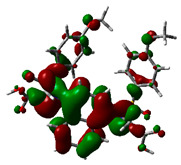	−8.5658	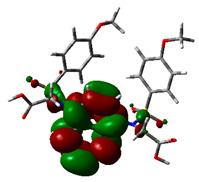	−5.9731	2.5927

**Table 5 molecules-28-04541-t005:** The results of threshold parameter values of the human hepatocellular carcinoma network analysis. ×10^−16^.

Gene	Betweenness	Closeness	Degree	Eccentricity	EigenVector	Radiality	Stress
SHC2	33	2.45	3	0.17	3.61E-16	28.74312	36
IGF1R	8	5.33	3	0.14	1.36E-16	29.22018	14
LEF1	22	2.00	3	1.01	9.65E-30	24.22936	22
CSNK1A1L	79.5	5.16	3	0.33	4.39E-30	25.68807	81
DVL1	67.5	4.60	3	0.14	2.53E-30	27.99083	70
PLCG1	32	3.95	3	0.17	−1.74E-17	28.23853	32
PRKCA	32	2.08	3	0.25	1.84E-15	27.2844	32
TERC	12.5	6.45	3	0.17	−3.22E-14	24.2844	13
EGFR	69	8.75	4	0.14	−3.04E-25	28.40367	69
HRAS	122	4.99	4	0.39	1.63E-30	54.9633	167
ARAF	122	4.06	4	0.55	2.11E-14	53.21101	172
GRB2	110	5.46	4	0.25	2.55E-15	57.21101	145
WNT16	81	10.03	4	0.04	1.23E-14	29.49541	102
RB1	16.5	2.00	4	2.00	−9.90E-30	47.02752	17
MET	92	7.28	4	0.04	−2.68E-15	29.10092	122
PIK3CA	61	5.08	5	0.5	−6.01E-15	57.19266	66
C05981	64	5.00	5	0.67	−1.71E-15	56.22936	68
NFE2L2	40	4.00	5	1.00	8.10E-15	25.58716	40
AKT3	57	5.5	6	1.00	−1.82E-15	53.80734	60
MAP2K1	97	9.5	10	2.00	6.27E-15	79.3945	100
TP53	21.5	8.28	8	0.20	−5.49E-14	23.59633	22
KEAP1	10.5	1.00	11	1.00	6.88E-31	29.38532	12

## Data Availability

The original contributions presented in the study are included in the article, and further inquiries can be directed to the corresponding author/s.

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
