# Peer review of "An In Silico Molecular Modelling-Based Prediction of Potential Keap1 Inhibitors from Hemidesmus indicus (L.) R.Br. against Oxidative-Stress-Induced Diseases"

_molecules, 2023, doi:10.3390/molecules28114541_

Round 1
Reviewer 1 Report
The article is informative. The authors used a wide range of computational methods to search for biologically active substances. The work is promising. When reading it, many questions arise, each of which can be the topic of a separate study. For example, questions regarding synergy and antagonism in the interaction of components with each other, and so on. However, these issues require separate studies and are beyond the scope of this work. In this regard, I do not ask them to the authors. Thus, realizing the scale of research and the problems of identifying natural compounds, I have few questions for the authors.
1. Almost all my remarks concern only the quality of Figures 8-11. In these figures, the numbers are almost invisible.
2. In addition, I believe that the results of the DFT method are superficially discussed in the work.
3. Is it possible for the authors to present the composition of the extracts in percentage terms? At least show the approximate composition of the predominant components. This information may also be of interest to readers.
Author Response
Dear Section Managing Editor,
Great thanks and appreciation for giving us the opportunity to submit a revised draft of the manuscript “An in silico molecular modelling-based prediction of potential Keap1 inhibitors from Hemidesmus indicus (L) R.Br. against oxidative stress-induced diseases” for publication in Molecules journal. We appreciate the time and effort you and the reviewers have devoted to providing comments on our manuscript and are grateful for the insightful comments and valuable improvements to our paper.
We've incorporated suggestions made by reviewers. These changes are highlighted red colour within the manuscript. Please see below, in red, for a point-by-point response to reviewers' comments and concerns.
Reviewer 1
The article is informative. The authors used a wide range of computational methods to search for biologically active substances. The work is promising. When reading it, many questions arise, each of which can be the topic of a separate study. For example, questions regarding synergy and antagonism in the interaction of components with each other, and so on. However, these issues require separate studies and are beyond the scope of this work. In this regard, I do not ask them to the authors. Thus, realizing the scale of research and the problems of identifying natural compounds, I have few questions for the authors.
Comment 1: Almost all my remarks concern only the quality of Figures 8-11. In these figures, the numbers are almost invisible.
Response 1: As per the suggestion, the quality of the images (Figure 8-11) enhanced in the revised manuscript.
Comment 2: In addition, I believe that the results of the DFT method are superficially discussed in the work.
Response 2: The complete and more details of DFT discussion included in our revised manuscript.
Comment 3: Is it possible for the authors to present the composition of the extracts in percentage terms? At least show the approximate composition of the predominant components. This information may also be of interest to readers.
Response 3: Thank you for your valuable comments, the presentage of individual components in the ethanolic extract of Hemidesmus indicus (L) R.Br was presented in our previous work, entitled “Optimization of Ultrasound-assisted extraction of bioactive compounds from Hemidesmus indicus (L) R.Br. using Response surface methodology and Adaptive neuro-fuzzy inference system” is accepted for publication in the journal of Food Science and biotechnology, Springer publisher. So that, we are not included in this article.

Reviewer 2 Report
The article written by Vellur et al., is about the the antioxidant potential of this plant extract was assessed by 21 antioxidant assays (DPPH, ABTS radical scavenging and FRAP). The authors used a total of 69 phytocompounds which were derived from this plant through the IMPPAT database, and their three-dimensional structures were obtained from the PubChem database. The chosen 69 phytocompounds were docked against the Kelch-Neh2 complex protein (PDB entry ID: 2flu, resolution 1.50 Å) along with the standard drug (CPUY192018). H. indicus (L) R.Br. extract (100 μg mL-1) showed 85 ± 2.917%, 78.783 ± 0.24% of DPPH, ABTS radicals scavenging activity, and 161±4 μg mol (Fe (II)) g−1 ferric ion reducing power. The top scored three hits were selected, namelyHemidescine (−11.30 Kcal mol−1), Beta-Amyrin (−10.00 Kcal × mol−1), and Quercetin (−9.80 Kcal mol−1) based on their binding affinities. Finally, they have also performed MD simulation studies to show protein-ligand stability. This manuscript has some serious concerns such as comparison with other similar systems available in the literature, and interpretation of results need further convincing explanation/arguments. However, this work may be recommended to publish after careful revision considering the below points and the revised version should be reviewed again before the publication.
1. In Table 1, the author must describe which phytochemical docked in which cavity and also for standard because if compounds are docked in different active sites, comparison is invalid.
2. Kindly give visualization of the grid with a box dimension in supporting information.
3. In section 2.5. Interpretation of Receptor–Ligand Interactions, kindly make table to compare interactions because it is very difficult to comprehend the jumbled text with similar residues again and again.
4. In Figure 4, a, c, e, g, the residue names were unreadable. Two things can be done to improve figure, dark blue protein surface can be reduced or manually names can be assigned, see some examples here in Chemical Physics Letters 767 (2021) 138379; Bioorg. Med. Chem. Lett. 43 (2021) 128079; 10.1515/znc-2021-0042.
5. After Molecular dynamics (MD) simulations, as the docked complexes changes so their intermolecular interaction and pose visualization is important.
6. What is significance of protein-ligand contacts timelines?
7. RMSF graphs should not be line graphs as they represents individual states see some examples here in Journal of Molecular Liquids 366, 120234, Molecular Simulation 48 (13), 1163-1174. Journal of Computational Biophysics and Chemistry 21 (02), 181-205
8. In MD simulations, what kind of solvation was adopted?
9. Can authors find out any relationship between HOMO-LUMO gap and reactivity of ligands?
The overall article is well-written still several typos can be corrected in review or proofs.
Author Response
Dear Section Managing Editor,
Great thanks and appreciation for giving us the opportunity to submit a revised draft of the manuscript “An in silico molecular modelling-based prediction of potential Keap1 inhibitors from Hemidesmus indicus (L) R.Br. against oxidative stress-induced diseases” for publication in Molecules journal. We appreciate the time and effort you and the reviewers have devoted to providing comments on our manuscript and are grateful for the insightful comments and valuable improvements to our paper.
We've incorporated suggestions made by reviewers. These changes are highlighted red colour within the manuscript. Please see below, in red, for a point-by-point response to reviewers' comments and concerns.
Reviewer 2
The article written by Vellur et al., is about the antioxidant potential of this plant extract was assessed by 21 antioxidant assays (DPPH, ABTS radical scavenging and FRAP). The authors used a total of 69 phytocompounds which were derived from this plant through the IMPPAT database, and their three-dimensional structures were obtained from the PubChem database. The chosen 69 phytocompounds were docked against the Kelch-Neh2 complex protein (PDB entry ID: 2flu, resolution 1.50 Å) along with the standard drug (CPUY192018). H. indicus (L) R.Br. extract (100 μg mL-1) showed 85 ± 2.917%, 78.783 ± 0.24% of DPPH, ABTS radicals scavenging activity, and 161±4 μg mol (Fe (II)) g−1 ferric ion reducing power. The top scored three hits were selected, namely Hemidescine (−11.30 Kcal mol−1), Beta-Amyrin (−10.00 Kcal × mol−1), and Quercetin (−9.80 Kcal mol−1) based on their binding affinities. Finally, they have also performed MD simulation studies to show protein-ligand stability. This manuscript has some serious concerns such as comparison with other similar systems available in the literature, and interpretation of results need further convincing explanation/arguments. However, this work may be recommended to publish after careful revision considering the below points and the revised version should be reviewed again before the publication.
Comment 1. In Table 1, the author must describe which phytochemical docked in which cavity and also for standard because if compounds are docked in different active sites, comparison is invalid.
Response 1: Based on the results obtained from Prank webserver, we have generated the grid box, the same was incorporated in our manuscript (2.3. Active Binding Site Identification) and the interacting amino acids of Kelch-Neh2 complex with test and standard drugs were given in 2.5. Interpretation of Receptor–Ligand Interactions and a supplementary table was attached with the manuscript.
Comment 2: Kindly give visualization of the grid with a box dimension in supporting information.
Response 2: We have added the grid box image in supplementary file
Comment 3: In section 2.5. Interpretation of Receptor–Ligand Interactions, kindly make table to compare interactions because it is very difficult to comprehend the jumbled text with similar residues again and again.
Response 3: Interpretation of Receptor–Ligand Interactions are now tabulated and presented in the supplementary table 1.
Comment 4. In Figure 4, a, c, e, g, the residue names were unreadable. Two things can be done to improve figure, dark blue protein surface can be reduced or manually names can be assigned, see some examples here in Chemical Physics Letters 767 (2021) 138379; Bioorg. Med. Chem. Lett. 43 (2021) 128079; 10.1515/znc-2021-0042.
Response 4: Thank you for the valuable suggestion, we have changed the color of protein surface and incorporated in revised manuscript.
Comment 5: After Molecular dynamics (MD) simulations, as the docked complexes changes so their intermolecular interaction and pose visualization is important.
Response 5: We have kept the images of docked complexes after Molecular dynamics (MD) simulations in the figures 8b, 9b,10b, 11b
Comment 6: What is significance of protein-ligand contacts timelines?
Response 6: Protein-ligand contacts timelines after MD simulation were shown in figures 8b, 9b,10b, 11b and the same was discussed in 2.8. Molecular dynamics (MD) simulation
Comment 7: RMSF graphs should not be line graphs as they represent individual states see some examples here in Journal of Molecular Liquids 366, 120234, Molecular Simulation 48 (13), 1163-1174. Journal of Computational Biophysics and Chemistry 21 (02), 181-205
Response 7: Thank you for the valuable suggestion. We have changed the RMSF graph.
Comment 8: In MD simulations, what kind of solvation was adopted?
Response 8: The TIP3P water solvation model was used to solve the docked protein-ligand complexes
Comment 9: Can authors find out any relationship between HOMO-LUMO gap and reactivity of ligands?
Response 9: The HOMO-LUMO gap used to finding the ligand stability, reactivity and electron conductivity. The HOMO-LUMO gap for the ligand is higher (eV), suggesting its high reactivity towards bonding with receptor. Furthermore, the HOMO energy shows that molecule is susceptible towards the electrophiles, whereas the LUMO energy indicates the susceptibility of the molecule towards the nucleophiles. As the energy gap increase the biological activity will increase.

Reviewer 3 Report
The study by Vellur et al. utilized molecular modeling to predict potential Keap1 inhibitors from Hemidesmus indicus (L) R.Br. for oxidative stress-induced diseases. The H. indicus (L) R.Br. extract exhibited antioxidant properties. Docking of 69 phytocompounds against the Kelch-Neh2 complex protein, along with a control drug, revealed the top three hits: Hemidescine, Beta-Amyrin, and Quercetin. These compounds showed favorable binding affinities and exhibited good molecular properties according to ADME/Tox predictions. MD simulations confirmed the stability of Keap1-ligand complexes. The manuscript is well-written and the findings could have potential implications in drug discovery for oxidative stress-related health complications.
Suggestions and concerns:
1. Figure 1. How many replicates were performed for the radical scavenging and FRAP assay experiments, please clarify.
2. Authors used CPUY192018, which is a potent Keap1 inhibitor. However, the original paper was not cited (J. Med. Chem. 2014, 57, 6, 2736–2745). In that paper, the CPUY192018 was docked into Keap1 structure. I would suggest that authors compare their CPUY192018-Keap1 complex model with the published one, which is clearly not consistent with each other.
3. Also, I noticed that the protonation state of CPUY192018 used is not correct, please check carefully and make corrections, as well as for other compounds. I would suggest preparing the ligands using software such as LigPrep, before performing molecular docking and molecular dynamics simulations studies.
4. For the ligand-Keap1 contact calculations, Fig 8-11, what is the cutoff value used for contacts? Please clarify.
Author Response
Dear Section Managing Editor,
Great thanks and appreciation for giving us the opportunity to submit a revised draft of the manuscript “An in silico molecular modelling-based prediction of potential Keap1 inhibitors from Hemidesmus indicus (L) R.Br. against oxidative stress-induced diseases” for publication in Molecules journal. We appreciate the time and effort you and the reviewers have devoted to providing comments on our manuscript and are grateful for the insightful comments and valuable improvements to our paper.
We've incorporated suggestions made by reviewers. These changes are highlighted red colour within the manuscript. Please see below, in red, for a point-by-point response to reviewers' comments and concerns.
Reviewer 3
The study by Vellur et al. utilized molecular modeling to predict potential Keap1 inhibitors from Hemidesmus indicus (L) R.Br. for oxidative stress-induced diseases. The H. indicus (L) R.Br. extract exhibited antioxidant properties. Docking of 69 phytocompounds against the Kelch-Neh2 complex protein, along with a control drug, revealed the top three hits: Hemidescine, Beta-Amyrin, and Quercetin. These compounds showed favorable binding affinities and exhibited good molecular properties according to ADME/Tox predictions. MD simulations confirmed the stability of Keap1-ligand complexes. The manuscript is well-written and the findings could have potential implications in drug discovery for oxidative stress-related health complications.
Suggestions and concerns:
Comment 1: Figure 1. How many replicates were performed for the radical scavenging and FRAP assay experiments, please clarify.
Response 1: 3 replicates
Comment 2: Authors used CPUY192018, which is a potent Keap1 inhibitor. However, the original paper was not cited (J. Med. Chem. 2014, 57, 6, 2736–2745). In that paper, the CPUY192018 was docked into Keap1 structure. I would suggest that authors compare their CPUY192018-Keap1 complex model with the published one, which is clearly not consistent with each other.
Response 2: Mentioned article cited in the revised manuscript
Comment 3: Also, I noticed that the protonation state of CPUY192018 used is not correct, please check carefully and make corrections, as well as for other compounds. I would suggest preparing the ligands using software such as LigPrep, before performing molecular docking and molecular dynamics simulations studies.
Response 3: Used Avogadro software for ligand preparation
Comment 4: For the ligand-Keap1 contact calculations, Fig 8-11, what is the cutoff value used for contacts? Please clarify.
Response 4: cutoff value 5Å

Round 2
Reviewer 2 Report
Regarding HOMO-LUMO energy gap, authors said, "Thus, a wide energy gap observed in the hit molecules negatively affect the electron to move from the HOMO to the LUMO, which subsequently led to a weak affinity of the inhibitor for Keap 1 protein.
In addition, HOMO-LUMO gap indicates Beta Amyrin and Hemidescine are the most reactive ligands."
Both statements are contrary to each other because Amyrin and Hemidescine have wide band gaps and these must be less reactive. Please consult some more literature. And its very rude that authors did not mention any literature in revision as suggested previously. Kindly go through the literature and correct these statements or avoid such statements.
No comments
Author Response
Reviewer 2
Comments and Suggestions for Authors
Regarding HOMO-LUMO energy gap, authors said, "Thus, a wide energy gap observed in the hit molecules negatively affect the electron to move from the HOMO to the LUMO, which subsequently led to a weak affinity of the inhibitor for Keap 1 protein.
In addition, HOMO-LUMO gap indicates Beta Amyrin and Hemidescine are the most reactive ligands."
Both statements are contrary to each other because Amyrin and Hemidescine have wide band gaps and these must be less reactive. Please consult some more literature. And its very rude that authors did not mention any literature in revision as suggested previously. Kindly go through the literature and correct these statements or avoid such statements.
Response:
In accordance with the comment, we corrected the first sentence by stating that: “Thus, a wide energy gap observed in the hit molecules negatively affect the electron to move from the HOMO to the LUMO, which subsequently led to a strong affinity of the inhibitor for Keap 1 protein.” Following the comment, we have included an appropriate citation.

Reviewer 3 Report
The authors addressed some of my comments satisfactorily, but the clarifications should be included in the manuscript. Comment #3 was not addressed adequately. The compound structure CPUY192018 in Figure 11 was incorrect, and it differed from the structure shown in Figure 4h. Similarly, there was an issue with the compound Hemidescine in Figure 8. I recommend that the authors carefully review these structures and simulations, and make the necessary corrections before publishing the results.
Author Response
Reviewer 3
Comments and Suggestions for Authors
The authors addressed some of my comments satisfactorily, but the clarifications should be included in the manuscript. Comment #3 was not addressed adequately. The compound structure CPUY192018 in Figure 11 was incorrect, and it differed from the structure shown in Figure 4h. Similarly, there was an issue with the compound Hemidescine in Figure 8. I recommend that the authors carefully review these structures and simulations, and make the necessary corrections before publishing the results.
Response: Thank you for your valuable comments for improvement of our manuscript. All the queries are incorporated in the revised manuscript. We have cross verified the images as suggested by you, we found that standard drug (CPUY192018 in Figure 11 and Figure 4h) was found to be correct. While Figure 8, due to technical error, it was wrongly presented, now we have replaced the correct image in the Figure 8 in our revised manuscript.

Round 3
Reviewer 3 Report
In the Figure 11, the sulfonyl groups of CPUY192018 should not connect to two hydroxyl groups. Also the carboxyl groups should not have two hydroxyl groups.
The correct structure of CPUY192018 should be
https://pubchem.ncbi.nlm.nih.gov/compound/73330369
Authors need to correct the structure, and re-run the simulations. Otherwise the results and conclusion should be wrong.
Author Response
Comments and Suggestions for Authors
In the Figure 11, the sulfonyl groups of CPUY192018 should not connect to two hydroxyl groups. Also the carboxyl groups should not have two hydroxyl groups.
The correct structure of CPUY192018 should be
https://pubchem.ncbi.nlm.nih.gov/compound/73330369
Authors need to correct the structure, and re-run the simulations. Otherwise the results and conclusion should be wrong.
Answer: Thank you for your valuable comments for improvements of our manuscript. As per your direction, we re-ran the molecular docking and molecular dynamics simulation of standard drug (CPUY192018) with correct structure (obtained from: https://pubchem.ncbi.nlm.nih.gov/compound/73330369). During the molecular docking studies there is no change in the docking score and some changes in the interactions between amino acids. The changes of receptor-ligand interactions were incorporated in the revised manuscript. All the obtained docking and dynamics images also incorporated.
Round 4
Reviewer 3 Report
The authors addressed my concerns well. No further concerns from me.